# Antibacterial Activity of a Promising Antibacterial Agent: 22-(4-(2-(4-Nitrophenyl-piperazin-1-yl)-acetyl)-piperazin-1-yl)-22-deoxypleuromutilin

**DOI:** 10.3390/molecules26123502

**Published:** 2021-06-08

**Authors:** Xiang-Yi Zuo, Hong Gao, Mei-Ling Gao, Zhen Jin, You-Zhi Tang

**Affiliations:** 1Guangdong Provincial Key Laboratory of Veterinary Pharmaceutics Development and Safety Evaluation, College of Veterinary Medicine, South China Agricultural University, Guangzhou 510642, China; zuoxiangyi@stu.scau.edu.cn (X.-Y.Z.); gh4264664@126.com (H.G.); gaomeiling90@gmail.com (M.-L.G.); jinzhenhami@scau.edu.cn (Z.J.); 2Guangdong Laboratory for Lingnan Modern Agriculture, Guangzhou 510642, China

**Keywords:** pleuromutilin derivative, antibacterial activity, pharmacokinetic, MRSA, LC-MS/MS

## Abstract

A novel pleuromutilin derivative, 22-(4-(2-(4-nitrophenyl-piperazin-1-yl)-acetyl)-piperazin-1-yl)-22-deoxypleuromutilin (NPDM), was synthesized in our laboratory and proved excellent antibacterial activity against methicillin-resistant *Staphylococcus aureus* (MRSA). In this study, more methods were used to further study its preliminary pharmacological effect. The antibacterial efficacy and toxicity of NPDM were evaluated using tiamulin as the reference drug. The in vitro antibacterial activity study showed that NPDM is a potent bactericidal agent against MRSA that induced time-dependent growth inhibition and a concentration-dependent post-antibiotic effect (PAE). Toxicity determination showed that the cytotoxicity of NPDM was slightly higher than that of tiamulin, but the acute oral toxicity study proved that NPDM was a low-toxic compound. In an in vivo antibacterial effect study, NPDM exhibited a better therapeutic effect than tiamulin against MRSA in a mouse thigh infection model as well as a mouse systemic infection model with neutropenia. The 50% effective dose (ED_50_) of NPDM in a *Galleria mellonella* infection model was 50.53 mg/kg. The pharmacokinetic properties of NPDM were also measured, which showed that NPDM was a rapid elimination drug in mice.

## 1. Introduction

*Staphylococcus aureus* is a potentially pathogenic Gram-positive bacteria that can cause infections ranging from mild (such as skin infection) to severe (such as postsurgical wound infections and systemic infections) [1]. *Staphylococcus aureus* possesses great ability to acquire antibiotic resistance. Only two years after the clinical use of methicillin, Jevons isolated MRSA in 1961 [2]. Nowadays, MRSA is almost universal and has become one of the most common pathogens in clinics [3]. Patients infected with MRSA suffer higher costs, longer hospital stays, and higher morbidity and mortality [4]. In bacteremia, for example, the in-hospital mortality rate for *S. aureus* bacteremia is nearly 40%, and the mortality rate of bacteremia patients caused by MRSA is 2.15 times higher than that caused by methicillin-sensitive *Staphylococcus aureus* (MSSA) [5]. Methicillin-resistant *S. aureus* is particularly resistant to β-lactam antibiotics, and this resistance is known to be caused by the penicillin-binding protein (encoded by genes designated 2’ or 2α) on the cell wall of *S. aureus*, which reduces the binding affinity between MRSA and β-lactam antibiotics [2]. It is urgent and necessary to develop new antibacterial agents that have different antimicrobial mechanisms against this microorganism.

Pleuromutilin (Figure 1), a natural compound, was isolated from *Pleurotus mutiliz* and *Pleurotus passeckeranius* by F. Kavanag in 1951 [6]. It displayed significant antibacterial activity against Gram-positive bacteria and mycoplasma [7]. The pleuromutilin derivatives selectively bind to the bacterial peptidyl transferase center and interfere with the synthesis of bacterial proteins [8,9]. This unique antibacterial mechanism reduces the cross-resistance between pleuromutilin and clinical antibiotics [10]. Nowadays, pleuromutilin has received increasing attention for the development of new antibiotics and the treatment of drug-resistant bacterial infections [11,12]. Lefamulin (Figure 1) is the first intravenous and oral pleuromutilin antibiotic approved by Food and Drug Administration, in 2019, for the treatment of community-acquired bacterial pneumonia [13]. As the first antibiotic with a new structure approved for systemic use in the past 20 years, lefamulin has aroused great interest in pleuromutilin among pharmacological researchers [14].

A series of pleuromutilin compounds were synthesized, and we performed a preliminary antimicrobial analysis. The results revealed that NPDM (Figure 1) possessed distinguished antibacterial effect against MRSA [15]. It motived us to further study its pharmacological characteristics including toxicity determination, in vitro and in vivo antibacterial activity, and pharmacokinetic studies.

## 2. Results

### 2.1. In Vitro Efficacy of NPDM

To evaluate the in vitro antibacterial efficacy, the minimum inhibitory concentrations (MICs) and minimum bactericidal concentrations (MBCs) of NPDM were determined against MRSA ATCC 43300, *S. aureus* ATCC 29213, and two clinical strains of *S. aureus* (AD_3_ and 144) using tiamulin as the reference drug. The results are shown in Table 1.

The results of time–kill assays are shown in Figure 2. The NPDM displayed bactericidal activity (i.e., ≥3 log_10_ CFU/mL decrease) against MRSA at the concentration of ≥2 × MIC after incubation for 24 h, while tiamulin showed bactericidal activity against MRSA at the concentration of ≥4 × MIC. Both NPDM and tiamulin showed a bacteriostatic effect on *S. aureus* when the drug concentration was no less than 2 × MIC.

To evaluate the in vitro antibacterial pharmacodynamics activity, the PAEs of NPDM and tiamulin against MRSA were then investigated. The results are shown in Figure 3 and Table 2.

### 2.2. Toxicity Determination

The cytotoxicity of NPDM and tiamulin were determined using the conventional 3-(4,5-dimethyl-2-thiazolyl)-2,5-diphenyl-2H-tetrazolium bromide (MTT) assay on normal rat liver cells (i.e., BRL-3A). The cell viability affected by different concentrations of the tested compounds is indicated in Figure 4. The 50% inhibition concentration (IC_50_) value of NPDM was 9.64 μg/mL, which was lower than that of tiamulin (13.1 μg/mL).

The results of the acute oral toxicity of NPDM are summarized in Table 3. After receiving a dose of 987.65 mg/kg of NPDM, only one mouse died, and five mice died within a day after receiving a dose of 5000 mg/kg of NPDM. The LD_50_ was 3764.49 mg/kg calculated by modified Karber’s formula, and the 95% confidence interval was 3006.67–4713.31 mg/kg. Swelling was found in the abdomen and back of some surviving and dead mice. For the dead mice, we also observed the rolling behavior and heard a strong gasping sound while they were alive.

### 2.3. In Vivo Effect of NPDM

In this study, the mouse thigh infection model and mouse systemic infection model were established to study the in vivo antibacterial activity of NPDM and tiamulin against MRSA or *S. aureus* ATCC 29213. The results are shown in Figure 5. The amount of inoculum in each thigh of neutropenic mice was 10^5~6^ CFU. After treatment with the test compound for 24 h, the average amount of MRSA inoculum in thighs of the NPDM-treated group, tiamulin-treated group, and untreated control group were (8.16 ± 3.70) × 10^7^ CFU, (2.64 ± 1.95) × 10^8^ CFU, and (3.22 ± 1.09) × 10^9^ CFU, respectively (Figure 5A), and that of the *S. aureus* were (1.23 ± 0.51) × 10^8^ CFU, (2.00 ± 1.21) × 10^8^ CFU, and (1.00 ± 0.79) × 10^9^ CFU, respectively (Figure 5B). Compared with the untreated group, treatment with NPDM (20 mg/kg) could significantly reduce mean MRSA counts (~1.61 log CFU/mL) in thighs (*p* < 0.05), indicating that NPDM possessed more potent in vivo antibacterial activity than tiamulin (decrease ~1.15 log CFU/mL). For *S. aureus*, treatment with NPDM at 20 mg/kg could significantly reduce mean bacterial counts (~0.95 log CFU/mL), which was more effective than tiamulin (~0.68 log CFU/mL).

We evaluated the therapeutic activity of NPDM and tiamulin using the mouse systemic infection model. The survival curves of NPDM and tiamulin at the dose of 30 mg/kg are shown in Figure 6. The untreated mice all died within 72 h following inoculation. Mice injected with sterile saline survived during the experiment. The results showed that the survival rate of NPDM against MRSA was 70% at the endpoint. This result illustrated that NPDM could protect mice from a lethal MRSA attack in vivo.

The *Galleria mellonella* infection model was utilized to investigate the ED_50_ of NPDM and tiamulin against MRSA. The survival rate of each group is shown in Figure 7. The dose of NPDM and tiamulin the led to the 50% effectiveness were 50.53 mg/kg and 75.74 mg/kg, respectively.

### 2.4. Pharmacokinetic Study

In order to assess the pharmacokinetic properties of NPDM, plasma samples were analyzed by LC-MS/MS. Representative LC-MS/MS chromatograms can be found in Appendix A in the Appendix A including chromatograms of blank plasma samples, blank plasma samples spiked with NPDM, and plasma samples taken 5 min after administration of NPDM. Under this LC-MS/MS condition, the retention time of NPDM was approximately 4.38 min. The results indicated that no interfering peak of endogenous substance was observed to affect the determination of NPDM. The concentration range of the calibration curve was 0~500 ng/mL. The typical equation and *R*^2^ value of the calibration curve were as follows: y = 386∙x + 52.1, *R*^2^ = 0.9998 (y: peak area of NPDM; x: concentration of NPDM in plasma). The limit of detection of NPDM was 0.1 ng/mL. The limit of quantification of NPDM was 2 ng/mL. The results revealed that the concentration of NPDM in plasma has linearity with its peak area in the range of 2~500 ng/mL. The extraction recovery ranged from 77.2~112%. The intra-day and inter-day precision ranged from 2.75% to 8.74% and 9.16% to 12.77% (see Appendix A.

The pharmacokinetic parameters of NPDM are shown in Table 4. The plasma concentration–time curve of NPDM after intravenous administration is displayed in Figure 8.

## 3. Discussion

In a previous study, we determined the MIC and MBC of NPDM and tiamulin against MRSA ATCC 43300 and *S. aureus* ATCC 29213 [15]. Considering that the high content of organic solvents may affect the value of MIC [16], in this study, we dissolved drugs using 5% DMSO, 5% Tween-80, and 90% normal saline rather than methanol. As shown in Table 1, NPDM showed excellent in vitro antibacterial activity against MRSA ATCC 43300, *S. aureus* ATCC 29213, AD_3_, and 144, and it was more potent than the reference drug tiamulin. The MBC/MIC value of NPDM against MRSA was one, which initially indicated that NPDM might possess great bactericidal effect.

The MIC and MBC are static measurements of antibacterial drugs [17]. To explore the profile of antibacterial activity over time, we conducted the time–kill test. As shown in Figure 2, the antibacterial effect of NPDM was positively correlated with time, not with the increase in drug concentration. These results demonstrate that NPDM induced time-dependent growth inhibition instead of dose-dependent growth inhibition. The NPDM displayed bactericidal activity against MRSA at a concentration of 2 × MIC, while tiamulin (2 × MIC) displayed bacteriostatic activity with a rebound growth. It is suggested that NPDM might be more beneficial for clinical use than tiamulin for its more significant bactericidal effect on MRSA.

The PAE was also investigated to evaluate the in vitro antibacterial pharmacodynamic activity of NPDM. The PAE refers to when the growth of bacteria remains under continuous inhibition after brief contact with antibiotics, even if the concentration of antibiotics drops below the minimum inhibitory concentration or disappears. From a clinical perspective, the PAE provides a theoretical basis for adjusting the dosing interval of some antibacterial agents and is of great significance for optimizing the treatment regimen and minimizing the adverse drug reactions [17,18]. At the concentration of 2 × MIC, the PAEs of NPDM after exposure for 1 h and 2 h were 2.58 h and 3.62 h, while at the concentration of 4 × MIC, they were 3.11 h and 4.03 h. The PAE was prolonged with the increase in the concentration of NPDM, which indicated that NPDM displayed a concentration-independent PAE (Figure 3). Additionally, the PAE of NPDM was 1~2 h longer than that of tiamulin (Table 2). It is suggested that NPDM might support longer dosing intervals than tiamulin when formulating a regimen for the treatment of MRSA.

The cytotoxicity study on BRL-3A illustrated that NPDM inhibited cell viability at low concentrations. The IC_50_ of NPDM was lower than that of tiamulin, which showed that the cytotoxicity of NPDM was slightly higher than that of tiamulin (Figure 4). Because the absorption of various drugs is different, the cytotoxicity may not fully reflect the toxicity of the drug. Therefore, acute toxicity experiments in vivo were conducted to study the toxicity of NPDM. An acute toxicity study identified the toxicity of NPDM and provided information on the maximum dose limit that can be used for a single feeding. The results showed that the LD_50_ of NPDM in mice was 3764.49 mg/kg, which was in the range of 501~5000 mg/kg. According to the classification of acute toxicity of an exogenous compound [19], NPDM was considered to be a low-toxic compound. The ATTM (Figure 1), the other semi-synthesis pleuromutilin derivative synthesized by Zhang Chao et al. [20], possesses great antibacterial effect on MRSA. The LD_50_ of ATTM in mice was 2304.4 mg/kg [20]. The oral acute toxicity of NPDM in mice was significantly lower than that of ATTM, which indicated that NPDM exhibits relatively high security and high potential for development as a new drug.

Since NPDM possessed potent antibacterial activity against MRSA in vitro and displayed high safety in the cytotoxicity study, we further utilized the mouse thigh infection model with neutropenia as well as the mouse systemic infection model to evaluate the in vivo antibacterial effect of NPDM against MRSA and *S. aureus*. Neutropenia mice are difficult to clear the pathogen through autoimmune response after infection. Thus, they can eliminate the interference of self-defense in pharmacodynamic studies [21]. As shown in Figure 5A, the number of bacteria in the thigh muscle of the two compound treatment groups were significantly decreased compared to the bacterial growth control group (*p* < 0.001), and the difference between the NPDM-treated group and the tiamulin-treated group was statistically significant (*p* < 0.05). The results indicated that both NPDM and tiamulin have great antibacterial effect on MRSA, and the antibacterial effect of NPDM was stronger than that of tiamulin. It can be seen from Figure 5B that *S. aureus* was less sensitive to NPDM than MRSA, which was consistent with the results of the in vitro time–kill study. The systemic infection model was utilized to determine the survival rate of mice after the MRSA lethal dose challenge. We found that the survival rate of the NPDM-treated group was 70% at the dose of 30 mg/kg, which was 20% higher than that of the tiamulin-treated group (Figure 6). This result indicated that NPDM possessed superior in vivo therapeutic activity to tiamulin against MRSA infection. We further studied the in vivo therapeutic efficacy of NPDM on the *Galleria mellonella* infection model. The *Galleria mellonella* infection model has the advantages of low cost and fast data collection and has been used as an alternative for mammalian models to investigate the in vivo efficacy of antibacterial drugs [22,23]. The ED_50_ of NPDM and tiamulin were 50.53 mg/kg and 75.74 mg/kg (Figure 7), which showed that NPDM displayed higher activity than tiamulin against MRSA in the *Galleria mellonella* infection model.

Inspired by the excellent antibacterial activity of NPDM against MRSA both in vivo and in vitro, the pharmacokinetic properties of NPDM were investigated to evaluate its potential for further research. After a single intravenous administration, NPDM displayed an acceptable plasma exposure level (AUC_0→__∞_ = 3.66 h·μg/mL). The T_1/2_ was 0.22 h, associated with high clearance (2.76 L/h/kg) and low MRT (0.36 h), which reflected that NPDM belongs to rapid elimination drugs (Table 4, Figure 8). To overcome the disadvantage of the short half-life of NPDM and make it more suitable for clinical application, we will make modifications that can prolong the half-life of NPDM such as using a sustained-release preparation or modifying its chemical structure.

## 4. Materials and Methods

### 4.1. Chemicals

The NPDM was synthesized according to our previous report [15]. The purity of NPDM was confirmed by RP-HPLC analysis. Tiamulin (Figure 1, purity: 98%) was purchased from Guangzhou Xiangbo Biotechnology Co., Ltd. (Guangzhou, China). All compounds were dissolved with 5% DMSO, 5% Tween-80, and 90% normal saline and filtered with a disposable 0.22 μm aseptic filter to prepare stock solutions. Stock solutions were stored at −20 °C.

### 4.2. Cell Culture and Preparation

The BRL-3A cells were purchased from the Shanghai Cell Bank of the Chinese Academy of Sciences (Shanghai, China). The cells were cultured in Dulbecco’s modified Eagle’s medium (DMEM) containing 10% FBS, 100 mg/mL streptomycin, and 100 IU penicillin at 37 °C with 5% CO_2_ and were sub-cultured every 2 or 3 days.

### 4.3. Bacterial Strains and Culture Medium

*Staphylococcus aureus* ATCC 29213 and MRSA ATCC 43300 were obtained from the Guangdong Microbial Culture Collection Center (Guangzhou, China). The *S. aureus* (AD_3_ and 144) were isolated from a pig farm in Guangdong Province. Brain heart infusion (BHI), mannitol salt agar (MSA), Mueller–Hinton agar (MHA), and Mueller–Hinton broth (MHB) were purchased from Qingdao Hope Bio-Technology Co. (Qingdao, China), Ltd. All bacterial strains were stored at −20 °C in 40% glycerol. A single colony was selected from the MSA plate, inoculated in BHI broth and pre-incubated at 37 °C to the logarithmic growth phase.

### 4.4. Animal

Adult specific pathogen-free ICR mice (22~25 g, 5~6 weeks) were purchased from Hunan Slack Jingda Experimental Animal Co., Ltd. (Changsha, China). All mice were adapted for 1 week before the beginning of the experiments. *Galleria mellonella* (*G. mellonella*), about 350 mg, were purchased from Tianjin Kaide Ruixin Co., Ltd., (Tianjin, China) and used within ten days. The protocol was approved by the Animal Research Committees of the South China Agriculture University.

### 4.5. Antimicrobial Activity Evaluation

#### 4.5.1. The MICs Assay

The MICs of NPDM against MRSA were determined by the broth dilution micro-method [24]. Briefly, the stock solution of NPDM and the reference drug tiamulin were diluted by a two-fold serial dilution method with MHB in a 96-well plate to provide concentration ranges of 16~0.032 μg/mL. Then 100 μL of bacterial suspensions in log phase were diluted to 10^6^ CFU/mL with MHB and were added to each well to make the final concentrations of the test compounds ranging from 8 to 0.016 μg/mL. The bacterial suspension (no drugs but solvent) was used as the positive control and MHB (no cultures) was used as the negative control. After incubation at 37 °C for 24 h, the MIC value was observed as the lowest drug concentration which inhibited the visible growth of bacteria in the test group. The MICs assay was carried out in triplicate.

#### 4.5.2. The MBCs Assay

Determination of MBCs refers to the minimum concentrations of antibacterial agents required to kill 99.9% of the test microorganisms after 24 h of incubation [24]. After obtaining the MICs value, the 96-well plates were placed at 37 °C and incubated for 24 h. One hundred microliter aliquots of the bacterial suspensions were taken from the wells without visible bacterial growth and then plated onto MHA plates. All plates were placed at 37 °C and incubated aerobically for 24 h for colony count, and the MBC was determined as the drug concentration corresponding to the number of colonies ≤ 5 on the culture medium. The MBC assay was conducted at least in triplicate.

#### 4.5.3. Time–Kill Assay

The bactericidal activity of NPDM was evaluated by the time–kill curve assay as described previously [25]. The MRSA and *S. aureus* in the logarithmic phase were diluted to 10^6^ CFU/mL with MHB. Then diverse concentrations of NPDM or tiamulin were added to the bacteria solution reaching the final concentration of 1 × MIC, 2 × MIC, 4 × MIC, and 8 × MIC, respectively. All bacterial inoculums were cultured at 37 °C with shaking. A total of 100 μL samples were taken from the subcultures at 0, 3, 6, 9, and 24 h, respectively, and continuously diluted 10-fold in sterile saline; then, 25 μL diluent was plated on the MHA plate. All plates were placed at 37 °C for 24 h. The total number of colonies on the plate was counted, and the results were expressed as log (CFU/mL).

#### 4.5.4. The PAE Assay

The PAE was carried out as described in the literature [26]. The MRSA was diluted to 10^6^ CFU/mL in MHB. The NPDM or tiamulin was added into the bacteria solution at the final concentrations of 2 × MIC and 4 × MIC. Then all tubes were incubated at 37 °C for 1 h or 2 h. Each of the above inoculums was diluted 1000-fold with pre-warmed MHB to remove the test compounds and placed in a water bath at 37 °C immediately. One hundred microliters of suspension were taken from each culture, diluted 10-fold in sterile saline, and inoculated on MHA plates at 0, 1, 2, 4, and 6 h after inoculation. The total number comprising the colonies was counted after incubation at 37 °C for 20 h. The PAE was calculated referring to published methods [26]. The experiment was performed in triplicate.

### 4.6. Toxicity Determination

#### 4.6.1. Cytotoxicity Assay

The cytotoxicity of NPDM was evaluated by the MTT assay as previously reported [25]. Briefly, the medium containing 1 × 10^5^ BRL-3A cells per 1 mL was seeded into a 96-well plate at 100 μL per well. After incubation for 48 h, media were replaced with 100 μL of serial dilutions of test compounds and incubated for 16 h. The absorbance (A) was measured at 490 nm by an automatic microplate reader (BIO-TEK Instrument Inc., Winooski, VT, USA). The cell viability was calculated according to Equation (1):(1)$$\mathrm{Cell}\;\mathrm{viability}\;(\%)\;=\;(A_{\mathrm{simple}}-A_{\mathrm{negative}})/(A_{\mathrm{positive}}-A_{\mathrm{negative}})\times 100\%$$

A graph was constructed by plotting the inhibition percentage versus various concentrations of NPDM and tiamulin, from which the IC_50_ of NPDM or tiamulin was calculated. The measurement was repeated in triplicate.

#### 4.6.2. Acute Oral Toxicity Evaluation

The acute oral toxicity of NPDM was evaluated according to the method reported in the literature [27]. Mice were divided into 6 groups (10 animals each group, half male and half female) and were fasted for 12 h before the experiment. The NPDM was dissolved in DMSO and suspended in 0.5% sodium carboxymethyl cellulose solution. The dosage levels of NPDM are shown in Table 3. The volumes of suspension by oral gavages were 0.1 mL/10 g. The vehicle control group received DMSO without drug. All animals were observed twice daily for symptoms and mortality for two weeks. At the end of the study, all surviving animals were sacrificed. The LD_50_ value and 95% confidence intervals of acute oral toxicity were calculated according to the equation in the literature [28].

### 4.7. Animal In Vivo Model

#### 4.7.1. Mouse Thigh Infection Model

The mouse thigh infection model was performed as described in the literature with minor modifications [29]. Briefly, female mice had induced neutropenia (neutrophil count ≤ 100/mm^3^) by administered intraperitoneally cyclophosphamide on day 4 (150 mg/kg) and day 1 (100 mg/kg) before the experiment. A bacterial suspension of 0.1 mL (MRSA or *S. aureus*) containing approximately 10^7^ CFU/mL microorganism was injected into the posterior thigh of mice to establish infection. Two hours after infection, mice were intravenously injected with NPDM or tiamulin at 20 mg/kg. After 24 h, thighs (six per group) were collected upon euthanasia. Then, thighs were weighed and homogenized with sterile saline. Tissue homogenate was continuously diluted 10-fold and plated on MHA plates for colony counts. SPSS 19 software was used for statistical analysis and *t*-tests.

#### 4.7.2. Mouse Systemic Infection Model

The survival rate of mice was determined by the systemic infection model according to the literature with some modifications [30]. Neutropenic mice (10 per group, half male and half female) were injected with 0.5 mL MRSA inoculum of 10^6^ CFU/mL (100% minimum lethal dose) intraperitoneally. One hour after infection, the mice were administered with NPDM or tiamulin intravenously at a dose of 30 mg/kg body weight. Tiamulin was used as a reference drug. The survival of the mice was recorded for 7 days.

#### 4.7.3. Galleria Mellonella Infection Model

The survival rate of *Galleria mellonella* was determined according to the literature with some modifications [22]. Larvae of *Galleria mellonella* (16 per group) were injected with 10 μL of 10^8^ CFU/mL MRSA inoculum into the last right proleg. Two hours after infection, larvae were injected with 10 μL of various doses (17.14, 25.71, 37.14, 57.14, and 85.71 mg/kg) of NPDM or tiamulin (used as reference drug) into the last left proleg. The larvae were maintained at 37 °C. Survival of the larvae was recorded daily for 5 days, and the ED_50_ was calculated.

### 4.8. Pharmacokinetic Studies

Adult female mice fasted for 12 h before the experiment. The NPDM was administered intravenously at the dose of 10 mg/kg. Approximately 0.2 mL blood samples were collected from the ocular venous plexus at 0.083, 0.167, 0.25, 0.5, 0.75, 1, 2, 4, 6, 8, 12, and 24 h after administration. All samples were centrifuged at 3000 rpm for 10 min, then the plasma samples were separated and stored at −20 °C. After thawing at room temperature, samples were mixed with acetonitrile (HPLC grade, sample:acetonitrile = 1:4, *v*/*v*) and centrifuged at 12,000 rpm for 10 min. Then, the supernatant was purified by the 0.22 μm cellulose membrane filter and transferred into an autosampler vial for HPLC-MS/MS analysis. Information of the equipment and analysis conditions are available in the Supplemental Materials. The pharmacokinetic parameters of the NPDM were calculated using the non-compartmental model in WinNonlin 5.2 software (Pharsight, Mountain View, CA, USA).

## 5. Conclusions

NPDM, a new derivative of pleuromutilin, was synthesized in our lab. Compared to tiamulin, NPDM displayed superior bactericidal effect against MRSA both in vitro and in vivo. Furthermore, toxicity studies showed that NPDM was a less-toxic antibacterial agent. The pharmacokinetic study showed that NPDM was a rapid elimination drug in mice. As the harm of multi-drug-resistant bacteria is becoming increasingly serious today, NPDM, given its effectiveness and safety demonstrated in this study, merits further studies as a novel antibacterial agent against MRSA infections.

## Figures and Tables

**Figure 1 molecules-26-03502-f001:**
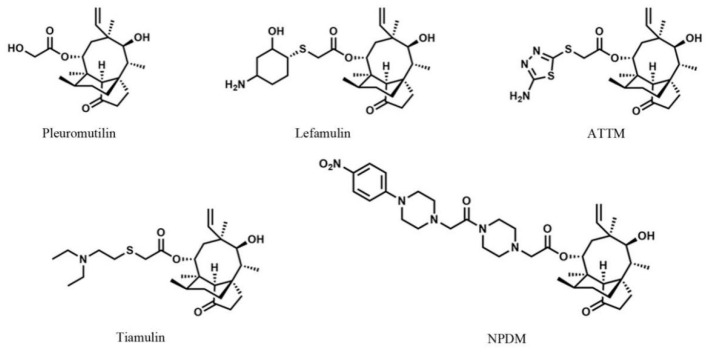
Structure of pleuromutilin, lefamulin, ATTM, tiamulin, and NPDM.

**Figure 2 molecules-26-03502-f002:**
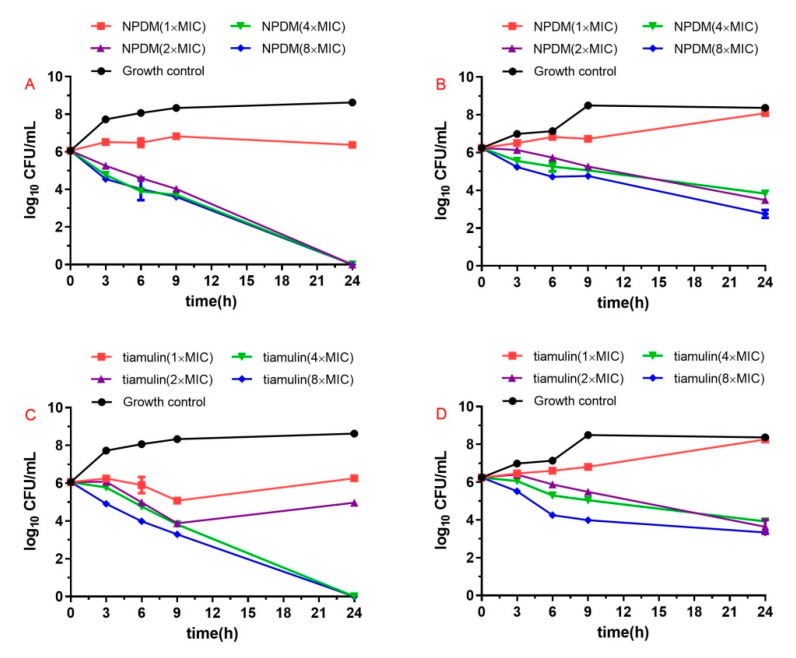
Time–kill curves for MRSA ATCC 43300 and *S. aureus* ATCC 29213 with different concentrations of NPDM and tiamulin: (**A**) NPDM against MRSA; (**B**) NPDM against *S. aureus*; (**C**) tiamulin against MRSA; (**D**) tiamulin against *S. aureus*.

**Figure 3 molecules-26-03502-f003:**
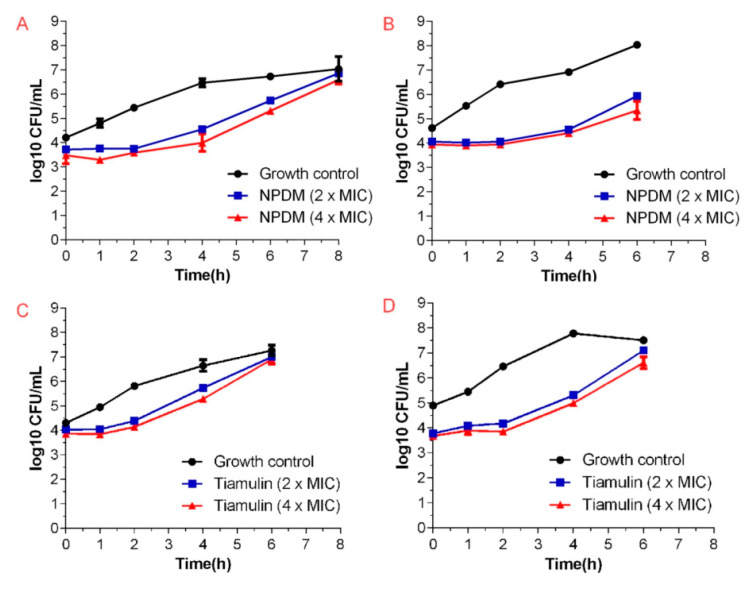
The bacterial growth kinetic curves for MRSA ATCC 43300 exposed to NPDM for 1 h (**A**) or 2 h (**B**) and tiamulin for 1 h (**C**) or 2 h (**D**).

**Figure 4 molecules-26-03502-f004:**
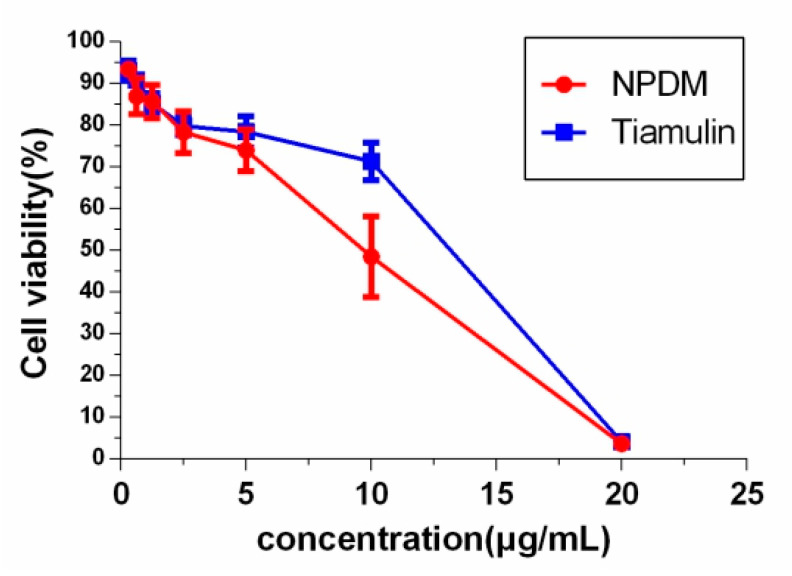
Cytotoxicity of NPDM and tiamulin using BRL-3A cells tested using the MTT methodology.

**Figure 5 molecules-26-03502-f005:**
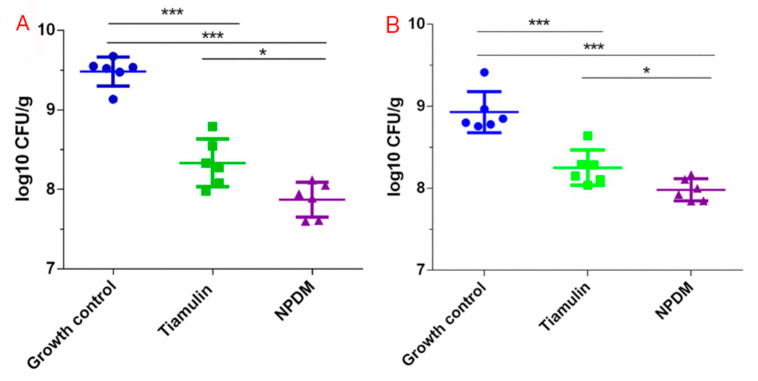
Efficacy of tiamulin (20 mg/kg) and NPDM (20 mg/kg) against MRSA ATCC 43300 in mouse neutropenic thigh models (**A**). NPDM vs. growth control, ***; tiamulin vs. growth control, ***; NPDM vs. tiamulin, *. Efficacy of tiamulin (20 mg/kg) and NPDM (20 mg/kg) against *S. aureus* ATCC 29213 in mouse neutropenic thigh models (**B**). NPDM vs. growth control, ***; tiamulin vs. growth control, ***; NPDM vs. tiamulin, *. (* 0.01 < *p* < 0.05; *** *p* < 0.001).

**Figure 6 molecules-26-03502-f006:**
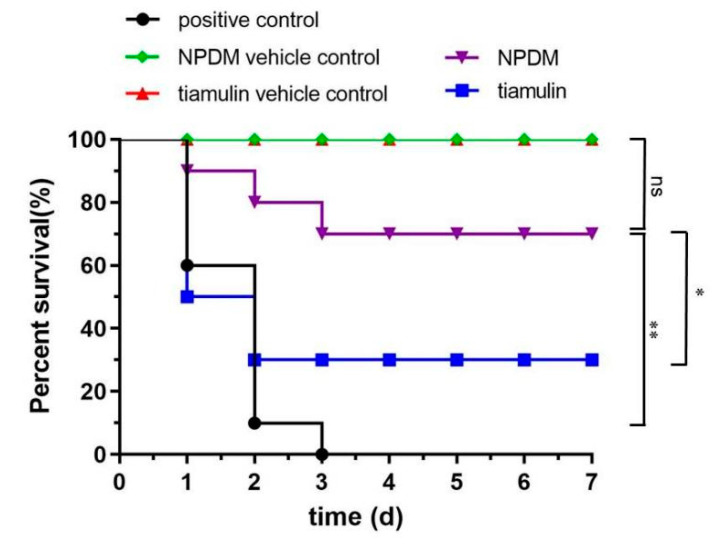
Survival of mice challenged with MRSA ATCC 43300 after treatment with 30 mg/kg body weight of NPDM (*n* = 10). NPDM vs. NPDM vehicle control, *ns*; NPDM vs. positive control, **; NPDM vs. tiamulin, *. (*ns*, not significant; * 0.01 < *p* < 0.05; ** 0.001 < *p* < 0.01).

**Figure 7 molecules-26-03502-f007:**
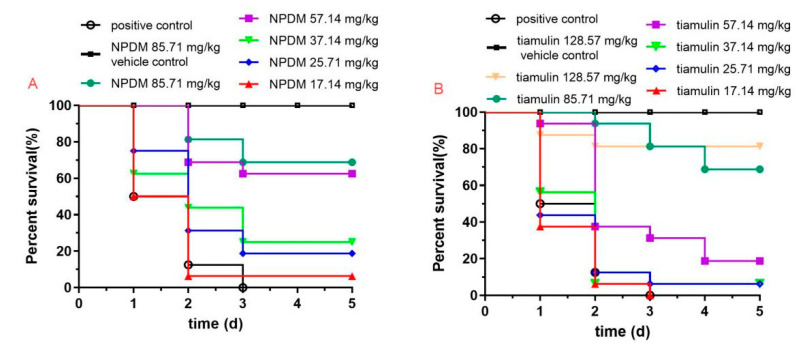
Survival of *G. mellonella* challenged with MRSA ATCC 43300 after treatment with different doses of NPDM (**A**) or tiamulin (**B**).

**Figure 8 molecules-26-03502-f008:**
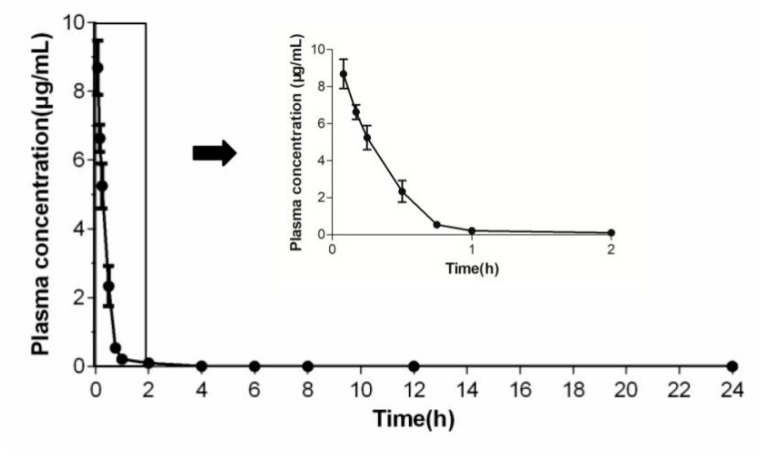
The plasma concentration–time curve of NPDM after intravenous administration.

**Table 1 molecules-26-03502-t001:** MIC values and MBC values (μg/mL) for NPDM and tiamulin against MRSA ATCC 43300, *S. aureus* ATCC 29213, and two clinical strains of *S. aureus* (AD3 and 144).

Compounds	MICs/MBCs (μg/mL)			
	ATCC 43300	ATCC 29213	AD_3_	144
NPDM	0.125/0.125	0.125/0.25	0.125/0.5	0.125/0.5
Tiamulin	0.5/0.5	0.5/0.5	0.5/2	0.5/2

**Table 2 molecules-26-03502-t002:** The PAE values of NPDM and tiamulin against MRSA ATCC 43300.

Compounds	Concentrations	PAE (h)	
		Exposure for 1 h	Exposure for 2 h
NPDM	2 × MIC	2.58	3.62
	4 × MIC	3.11	4.03
Tiamulin	2 × MIC	1.53	1.65
	4 × MIC	1.90	2.04

**Table 3 molecules-26-03502-t003:** Oral single-dose toxicity of NPDM in mice.

Group	*n*	Dose (mg/kg b.w.)	Logarithmic Dose	Mortality	Mortality Rate (%)
1	10	987.65	3.70	1	10
2	10	1481.48	3.52	1	10
3	10	2222.22	3.35	2	20
4	10	3333.33	3.17	3	30
5	10	5000	2.99	5	50
DMSO	10	0.1 mL/10 g b.w.	-	0	0

**Table 4 molecules-26-03502-t004:** The non-compartmental parameters of NPDM. (C_max_, maximum concentration; T_max_, time to reach C_max_; T_1/2_, half-life; CL, clearance; MRT, mean resident time; AUC_0→t_, area under the curve from zero to the last measurable plasma concentration; AUC_0→__∞_, area under the curve from the last measurable plasma concentration to infinity).

Parameters	Mean ± SD
C_max_ (μg/mL)	8.69 ± 0.74
T_max_ (h)	0.083
T_1/2_ (h)	0.22 ± 0.04
Cl (L/h∙kg)	2.76 ± 0.31
MRT (h)	0.31 ± 0.05
AUC_0-t_ (h·μg/mL)	3.65 ± 0.40
AUC_0-∞_ (h·μg/mL)	3.66 ± 0.37

## Data Availability

Not available.

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
