# Peer review of "Antibacterial Activity of a Promising Antibacterial Agent: 22-(4-(2-(4-Nitrophenyl-piperazin-1-yl)-acetyl)-piperazin-1-yl)-22-deoxypleuromutilin"

_molecules, 2021, doi:10.3390/molecules26123502_

Round 1

Reviewer 1 Report

The contribution by Zuo et al., describes the effect of a pleuromutilin derivative on several Gram-positive bacteria, most importantly methicillin-resistant Staphylococcus aureus ATCC 43300 (MRSA). The derivative, 22-(4-(2-(4-nitrophenyl-piperazin-1-yl)-acetyl)-piperazin-1-yl)-22-deoxypleuromutilin (NPDM), has been prepared and evaluated earlier by the same group (Gao et al., DOI: 10.1016/j.ejmech.2017.01.004). In this contribution, the authors investigate the effects of NPDM in more detail.

This contribution is based on sound experiments. I do, however, have some concerns with how the results are presented.

Major remarks:

  1. in the part where minimum inhibitory concentration (MIC) and minimum bactericidal concentration (MBC) have been determined for NPDM and tiamulin against MRSA (ATCC 43300) and aureus (ATCC 29213) (all lines 64-67 and Table 1) the authors should clearly indicate that the MIC values have been determined before for both compounds and both strains (Table 1 in Gao et al., DOI: 10.1016/j.ejmech.2017.01.004).
  2. the authors should point out that the MIC value for NPDM published in Gao et al. is in fact different from the MIC value presented here in Table 1 (0.015 vs 0.125 ug/ml). I would suggest that the authors also provide an explanation in the main text.
  3. if the MIC and MBC experiments have been indeed done in triplicate (Methods, line 271 and 280) Table 1 should have the values presented as means and standard deviation should be added.
  4. It should be specified that the MBC values were determined in the 24 hour window.
  5. the number of replicates is not indicated for the post-antibiotic effect (PAE).
  6. In Results, it should be specified what an MTT assay is and what kind of cell line is BRL-3A (lines 88-91). In general, abbreviations should be spelled out the first time they appear in the main text.
  7. since tiamulin is used as a reference throughout the manuscript, I would be interested to know what the acute oral toxicity is for tiamulin (lines 94-100). I am not suggesting additional experiments if these values are not known, but if they are, they should be specified here.

Minor remarks:

  1. it is not necessary to provide the genus abbreviation (in parentheses, line 27). It should be abbreviated, however, in the following text (e.g. line 36).
  2. lines 37-38 “protein-binding protein” is likely a mistake. The authors are probably referring to the penicillin-binding protein 2a.
  3. “gram” in Gram-positive/negative should be capitalized (e.g. line 44).
  4. line 45, abbreviation PTC is not used in the text and is therefore redundant. Similarly, abbreviation FDA and CABP are not used either (line 51).
  5. line 64: MIC and MBC abbreviations should be spelled out.
  6. line 80: in vitro should be italicized.
  7. line 118: tiamulin is unintentionally capitalized.
  8. sample size should be indicated in Figure 6 legend.
  9. lines 135-136: this sentence should be rearranged; it is not clear in the current form.
  10. line 240: please specify that the cells are fibroblasts.
  11. References are not uniformly formatted (title capitalization, some formatting missing, etc.)

Reviewer 2 Report

The manuscript entitled “Antibacterial activity of a promising antibacterial agent: 22-(4-(2-(4-Nitrophenyl-piperazin-1-yl)-acetyl)-piperazin-1-yl)-22-deoxypleuromutilin”, the authors investigated for preliminary pharmacological effect of pleuromutilin derivative, NPDM.

The results (antimicrobial activity, toxicity, animal in vivo and pharmacokinetic studies) in this study are helpful for developing novel antibacterial agents.

The manuscript is clearly written and well analyzed. Therefore, the manuscript is acceptable for this journal. 

Reviewer 3 Report

The manuscript by Tang and coworkers reports further biological/pharmacological characterizations of a pleuromutilin derivative previously synthesized and studied in the research group (ref. 15). In general, the manuscript is well-written and the conclusions are supported by the experimental data reported. This reviewer recommend publication, after a few points are addressed, as listed below:

The results described partially overlap those reported in ref. 15 [10.1016/j.ejmech.2017.01.004] and this reduces the novelty of the manuscript. For instance, Table 1 reports the antibacterial activity of compound NPDM (formerly called 11c) and tiamulin (reference compound) against two MRSA strains. Table 1 of ref. 15 reports exactly the same assay. To improve the novelty of the manuscript and to account for repetition of the results in two papers, the authors should add a few lines in the discussion, on the reasons why the two "Table 1" (namely, the present one and that in ref. 15) report quite different MIC values for compound NPDM/11c against the same bacterial strains, e.g., 0.015 vs. 0.125 ug/mL, against ATCC43300. Curiously, also the reference compound tiamulin was found less active, although not tenfold as NPDM).

Similarly, the two papers (the present one and ref. 15) report the same in vivo test, but - in the present paper - with a tenfold decrease in the amount of CFU/mL of the inoculum, compared to a drop in the compound concentration of only 10 mg/Kg. In ref. 15, the authors state that: "The neutropenic animals (10 per group) were inoculated intraperitoneally with 0.5 mL of an inoculum containing ∼107 CFU/mL MRSA. Mice were then administered with the test compounds intravenously 1 h after infection at dose of 40 mg/kg.", while, in the present work: "Neutropenic mice (10 per group, half male and half female) were injected with 0.5 mL MRSA inoculum of 106 CFU/ mL intraperitoneally. One hour after infection, the mice were administered with NPDM or tiamulin intravenously at a dose of 30 mg/kg body weight."

To account for almost the same test reported twice, the authors should discuss why they decide to use a tenfold lower amount of inoculum and what conclusion can be drawn by the new assay, compared to the one already published in 2017.

Minor points:

- p.1, Introduction, 8 lines from the top: "S. aureus" (italics) (instead of staphylococcus aureus). It seems that a few italics were lost in formatting the paper. Please, check throughout text also terms such as "in vitro" (e.g., line 80) and "in vivo".

- Table 1, second column: please, replace "MRSA" with "ATCC 43300".

- p. 3: please improve image quality for Figure 2.

- Throughout text: "tiamulin" (initial not capitalized).

- p.6, Figure 7: it is not possible to follow the lines in panel B. Please, either reduce the line thickness or move the lines to avoid overlapping.

- Figures 6 and 7: please, rename the negative controls to clearly state that they are negative controls. The presence of the tested compound (or concentration) in the label is misleading.

- Throughout text, please do not use the past when referring to text Tables and Figures (e.g., "The pharmacokinetic parameters of NPDM were shown in Table 4" should read: "The pharmacokinetic parameters of NPDM are shown in Table 4").

- p.7, last line (194): please add a reference to "Zhang Chao et al.". By the way, please rewrite the sentence: "ATTM (Figure 1), the other semisynthesis pleuromutilin derivative was synthesized by Zhang Chao et al., possesses great antibacterial" activity.

Reviewer 4 Report

Summary: This manuscript demonstrates the potential of NPDM as an antibiotic and the authors did a good job of testing several different parameters—in vitro activity, cytotoxicity, in vivo efficacy.  Overall, the experiments were well-executed and clear, although I will suggest a couple additional experiments to better clarify NPDM activity.  There were also some issues with grammar and syntax that the authors will need to correct. Specific comments are below.

  1. The authors show that NPDM has bactericidal activity in figure 2. The authors never state this explicitly, but reading their methods, I assume that bacteria either were in log phase or could have reentered log phase upon dilution into media at the beginning of the assay.  The authors first should clarify this but second, they should also test bactericidal activity against both log and stationary phase S. aureus as bactericidal activity can vary significantly depending on growth phase.  The authors should also insure that the bacteria remain in stationary phase throughout the time kill assay—i.e. add the antibiotics to cultures already in stationary phase rather than diluting stationary phase into fresh media with antibiotics.  Lastly, the authors do not remove the antibiotic before dilution plating.  Although they are diluting the antibiotic during serial dilutions and the antibiotic may be so dilute it no longer has bactericidal activity, the authors should not rely on this assumption as it would mean artificially low colony counts. Either they should test that the diluted antibiotic can no longer contribute to bacterial killing or they should remove the antibiotic via centrifugation and media replacement prior to serial dilution and plating.
  2. The authors should try their MIC assays in the presence of serum, since this is often inhibitory towards antibiotics. Given that the antibiotic did work in vivo, the serum cannot completely inactivate NPDM, although it could still inhibit its activity.  Human serum is ideal, however, if that is not available animal serum could also be used.
  3. The authors have a tendency to refer to their data in the past tense. For example, line 66: “The results were shown”.  This should be present tense—The results are shown in Table 1—since the results are displayed in present time.  This occurs several places and should be corrected.
  4. The authors do not always introduce the logic behind their experiments in their results and this makes it very choppy to read. For example, it would have been helpful to give some explanation of what a PAE is and why the authors are testing this as they introduce the data in the results.  Same with when the authors introduce the in vivo effect of NPDM in section 2.3.  It felt like they were starting that section with what should have been the second or third sentence in the paragraph.  Why did the authors choose a neutropenic model?  What was the goal with starting this model?  Section 2.4 would also benefit from some sort of opening sentence, even just “In order to assess the half-life of NPDM, plasma samples were analyzed by LC-MS/MS”.
  5. The statistical tests the authors used in figure 5 should be included in the figure legend.
  6. Figure 6: Do “Tiamulin control” and “NPDM control” refer to vehicle control? If so, state Tiamulin vehicle control or “saline control/DMSO control”. That is clearer. Also in figure 6, the authors should perform a statistical analysis of the survival curves. Keep in mind the following when making multiple comparisons of survival curves: https://www.graphpad.com/guides/prism/latest/statistics/stat_multiple_comparisons_of_surviv.htm.

Minor Grammatical issues:  there are several examples of grammatical errors or awkward sentence phrasing.  I have listed some suggested changes below, but the manuscript should be edited for proper grammar as this is not an exhaustive list.

Line 36: “MRSA is particularly resistant to beta-lactam antibiotics….”

Line 57/58: “…have been synthesized and our group has performed preliminary antimicrobial analysis.”

Line 185: “Because the absorption of various drugs is different, the cytotoxicity may not fully reflect…

Line 198-199: This opening sentence in the paragraph is very awkward and should be completely reworded.  It might help to break this into two sentences.

Lines 188-189: The two paragraphs about toxicity can be combined into one.

Line 214-215.  The paragraph with the Galleria results can be combined with the previous paragraph.  I suggest getting rid of the “Then” that opens line 215 and just starting the sentence with “We further studied…”

Line 228-230: “….more suitable for clinical application, we will make modifications that can prolong the half-life of NPDM such as using a sustained-release preparation or modifying its chemical structure.”

Line 265: “…bacterial suspensions were diluted….”

Round 2

Reviewer 4 Report

The authors addressed some but not all of my concerns in their revision.  Specific comments are below:

  1. The authors claim bactericidal activity but ignored my suggestion that they should also test stationary phase bacteria so that they know whether this activity is bactericidal against both stationary and log phase or only log phase. This is reasonable to assess since antibiotics are more likely to have bactericidal activity against actively dividing bacteria, however, it is not uncommon for an infection to have slowly dividing bacteria and hence be more similar to stationary phase.  They must either 1) actually test NPDM activity against stationary phase bacteria or 2) clarify that NPDM is “bactericidal against log phase bacteria” when they mention the bactericidal activity so it is clear to future readers what they actually tested and the limitations of their assessment.

  1. The authors state they have performed control experiments to make sure that not removing antibiotic prior to dilution plating did not impact bacterial growth. That information should be included as data not shown somewhere in their manuscript, either in their methods or results.

  1. The authors missed the point of my suggestion that they assess the effect of serum in their MICs. Their MICs without serum are valid; that is the standard way to determine an MIC. My question is whether the inclusion of serum impacts the MIC as components of serum can inactivate antibiotics.  This can be a challenge in antibiotic development and therefore it is very common for MIC to be assessed both with and without serum (i.e. 5-10% FBS). I see no reason why a MIC assay must be delayed to future studies when this is a very simple experiment and something normally done very early in antibiotic assessment—exactly the type of experiments the authors are currently doing in this manuscript.

  1. Figure 5: The authors still have not included the statistical tests they used in this figure. They have defined what p-values their asterisks represent but what test did they use to obtain those p-values? It is unnecessary to include “NPDM vs growth control, ****; tiamulin vs. growth control, ***”in the figure legend.  That information is contained in the figure. 

Here is a suggestion for the figure legend of figure 5:

Figure 5. Efficacy of tiamulin (20 mg/kg) and NPDM (20 mg/kg) in a neutropenic thigh mouse model against (A) MRSA ATCC 43300 or (B) S. aureus ATCC 29213.* 0.01 < p < 0.05; ** 0.001 < p < 0.01; *** p < 0.001) as determined by ___________________ (place the name of the statistical test you used to determine the p-values in the blank).

  1. Figure 6: The authors state the results in the figure legend as well as displaying them graphically. This is not necessary.  Instead of stating “positive control” the authors should simply state “Untreated mice”.  It should also be noted somewhere that the vehicle control animals were not infected with MRSA—at least I assume that is why they have 100% survival. The authors state that they use the vehicle control for both compounds but do not state in their methods what those vehicle controls are. 

The authors are making multiple comparisons on their survival curves in figure 6--did they compute the Bonferroni corrected threshold for each individual comparison to account for the error that comes from making multiple comparisons? If not, how did they account for this?  It is not explained in your methods or in your legend.  Also, once again, the authors fail to state what statistical test is used to generate their p-values. 

Grammatical issues:

Line 77:  incubation for 24h, while tiamulin (the authors have a period instead of a comma)

Line 79: "..when the drug concentration was less than 2X the MIC."

Line 111: “…were used to study the in vivo…”

Line 182: “….pure methanol.”

Line 229-230: The sentences below are poorly constructed and must be reworded.

“Neutropenia mice are difficult to clear the pathogen through autoimmune response after infection. So they can eliminate the interference of self-defense in pharmacodynamic studies [21].